# LLM Program Optimization via Retrieval Augmented Search

## Abstract

With the advent of large language models (LLMs), there has been great interest in applying them to solve difficult programming tasks. Recent work has demonstrated their potential at program optimization, a key challenge in programming languages research. We propose a blackbox adaptation method called Retrieval Augmented Search (RAS) that performs beam search over candidate optimizations; at each step, it retrieves in-context examples from a given training dataset of slow-fast program pairs to guide the LLM. Critically, we find that performing contextual retrieval based on an LLM-generated natural language description significantly outperforms retrieval based on the source code. In addition, we propose a method called Aegis for improving interpretability by decomposing training examples into "atomic edits" that are significantly more incremental in nature. We show that RAS performs up to $2.04\times$ better than prior state-of-the-art blackbox adaptation strategies on optimizing C++ programs, and that Aegis performs $1.37\times$ better while performing significantly smaller edits. We also show that using RAS improves the mean runtime percentile of Python programs by 10.27 as compared to other strategies.

## 1 Introduction

Given the success of large language models (LLMs) in writing code, there has been significant interest applying them to programming tasks. A particularly interesting task is program optimization, a long-standing problem in programming languages. Recent work has shown that LLMs have difficulty with this task out-of-the-box (Shypula et al., 2024)—intuitively, data on program performance is simply not widely available in traditional training datasets, making adaptation necessary.

To address this problem, they propose the "Performance Improving Edits (PIE)" benchmark, and use it to test a number of carefully designed adaptation strategies to identify effective algorithms for improving performance, including *blackbox* (i.e. prompting-based) adaptation strategies such as instruction prompting (Mishra et al., 2022), in-context learning (Brown et al., 2020), chain-of-thought prompting (Wei et al., 2022), and retrieval augmented generation (Lewis et al., 2020). They find *dynamic code retrieval* to be most effective; this approach retrieves a handful of slow-fast program pair examples from the training set at test time that are most relevant to the current instance (based on embedding similarity). These pairs are then used as in-context examples to prompt the LLM. Intuitively, this approach makes effective use of the training set, which contributes to its success.

This existing approach is "end-to-end" in the sense that it takes an input program and asks an LLM to directly output an optimized version of that program. However, this strategy differs significantly from how modern compilers work. Rather than making edits inspired by a handful of end-to-end examples, they systematically modify the program through a series of *compiler passes*, each of which is designed to perform a specific kind of optimization. These optimizations are inspired by existing examples, but in a way that generalizes them so they apply to new programs. Thus, a natural question is whether breaking end-to-end optimization into more incremental steps can improve performance.

Inspired by modern compiler design, we propose two novel retrieval-based adaptation strategies. First, we propose *retrieval augmented search* (RAS), which combines two insights to improve dynamic retrieval. First, rather than retrieve based on the code itself, it uses *contextual retrieval*, where it retrieves examples from the training set based on an LLM-generated natural language description of the program, abstracting the core algorithms and data structures used by the program from how they are implemented on a superficial level. Second, rather than retrieve a fixed set of programs,

we perform beam search by iteratively performing the retrieve-optimize-evaluate loop. These two techniques result in a state-of-the-art blackbox technique for adapting LLMs to program optimization.

However, this technique still produces large changes that can be hard to interpret. To further address this issue, we propose *Atomic Edit GuIded Search* (AEGIS), which leverages a preprocessing step to distill generalizable insights from the training data. In particular, we prompt the LLM to decompose a single slow-fast program pair in the training set into a sequence of *atomic edits*, which are incremental modifications associated with a natural language description of the edit, and then explain why the edit might improve performance. The description is intended to be generalizable, abstracting away specifics of the training example from which they are derived. After generating a dataset of atomic edits and examples associated with each edit, when given a new program, we use RAS to first search over incremental edits to this program. Each edit to this program is achieved by retrieving the most relevant atomic edit in our database and then prompting the LLM to apply this atomic edit to the new program. We then perform beam search over sequences of incremental edits to select the resulting program that achieves the greatest performance gain while preserving correctness.

We evaluate our approach using the PIE benchmark (Shypula et al., 2024) for C++ program optimization and using the Mercury benchmark (Du et al., 2024) for Python program optimization. On PIE, we show that RAS significantly outperforms dynamic retrieval, a state-of-the-art blackbox adaptation strategy, achieving an $8.61\times$ average speedup compared to $4.23\times$ for dynamic retrieval using Qwen3-Coder. Furthermore, AEGIS achieves a $6.08\times$ average speedup using GPT-4o, while reducing the average edit size (measured by string edit distance) by 17% when compared to RAS (with GPT-4o) and by 30% when restricting to the first edit in the search process (which is the most substantial one). Hence, RAS performs upto $2.04\times$ better than dynamic retrieval, while AEGIS performs $1.37\times$ better. We then show that by executing RAS on Mercury and comparing against our best-performing baselines, we can improve the mean runtime percentile by 10.27 for Qwen2.5-7B, significantly narrowing its performance gap as compared to more recent reasoning models. These results demonstrate that RAS and AEGIS are promising strategies for blackbox adaptation of LLMs to code optimization.

**Related work.** Code optimization has long been a problem of interest in programming languages. However, these approaches typically operate at a lower level of abstraction and are incapable of producing high-level optimizations such as changing algorithms and data structures. Thus, there has been recent interest in leveraging LLMs to augment existing, symbolic techniques. One approach that directly uses LLMs to perform program optimization is the Search-Based LLM (SBLLM) (Gao et al., 2024), which proposes an evolutionary search framework to iteratively optimize Python and C++ programs. However, in their framework, retrieval and search are not integrated, and they do not use contextual retrieval. Furthermore, they only report speedups of $1.55\times$ on the PIE benchmark (using GPT-4), so even the existing dynamic retrieval approach studied in PIE substantially outperforms their approach. Finally, Qiu et al. (2025) studies capabilities of LLMs for Python program optimization, finding significant gaps compared to human experts. We focus on optimizing C++ code since performance can be measured in a reproducible way using a simulator (Shypula et al., 2024).

Retrieval augmented generation is broadly known to improve code generation (Wang et al., 2024). The specific idea of dynamically retrieving relevant in-context examples from a larger training set was first proposed in Poesia et al. (2022) and was later shown to be highly effective for program optimization (Shypula et al., 2024). Recently, MapCoder (Islam et al., 2024) has shown that retrieving "previously seen" programming examples can improve code generation on the HumanEval benchmark. While contextual retrieval has recently been popularized for LLMs (Anthropic, 2024), the idea of annotating code to improve code search has long been studied extensively in software engineering. Older techniques such as Portfolio (McMillan et al., 2011) rely on information retrieval methods such as PageRank. The idea of automatically generating the natural descriptions for code snippets artificially was proposed in CoaCor (Yao et al., 2019), which trains a bidirectional LSTM to generate natural language descriptions optimized for use by a retrieval model.

## 2 RETRIEVAL AUGMENTED SEARCH

We describe our retrieval augmented search (RAS) algorithm (summary in Figure 1 and Algorithm 1).

---

**Algorithm 1** Retrieval Augmented Search (RAS)

---

**input:** $p_0, \Pi_{\text{train}}, F_{\text{opt}}, F_{\text{context}}, R, \phi$
**for** $i \in [1, ..., m]$ **do**
  $\Pi_i \leftarrow \text{top-}k\{((p, p'), d_\phi(p_{i-1}, p)) \mid (p, p') \in \Pi_{\text{train}}\}$
  $p_i^j \sim F_{\text{opt}}(\pi_i^j, p_{i-1}) \ (\forall j \in [k])$                    $\triangleright \Pi_i = \{\pi_i^j\}_{j=1}^k$
  $p_i \leftarrow \arg\max_{j \in [k]} R(p_i^j)$
**return** $p_m$

---

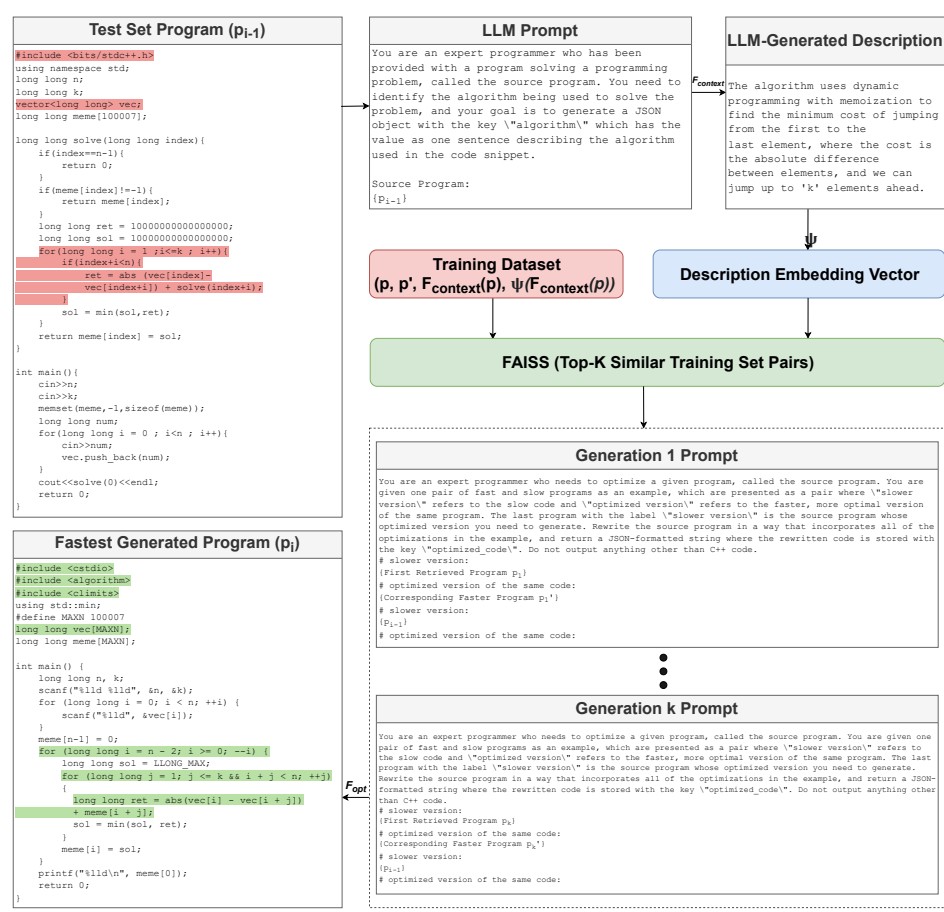

Figure 1: **RAS Framework**: For a given slow program $p_{i-1}$, we use $F_{\text{context}}$ to generate a program description and $\Psi$ to generate its corresponding description embedding vector. We retrieve similar training set programs using FAISS and pass them to $F_{\text{opt}}$. The fastest program generated by $F_{\text{opt}}$ is $p_i$.

**Problem formulation.** In the program optimization problem, the goal is to take a program $p \in \mathcal{P}$ as input, and output an optimized program $p' \in \mathcal{P}$ that is semantically equivalent to $p$. Typically, we are additionally given a set of test cases $\{(x_i, y_i)\}_{i=1}^k$ to check correctness; then, denoting the output of program $p$ on input $x$ as $p(x)$, we are searching for programs $p$ such that $p(x_i) = y_i$ for all $i \in \{1, ..., k\}$. While test cases do not guarantee semantic equivalence, they are widely used in machine learning for checking program equivalence (Chen et al., 2021; Li et al., 2022).

For PIE, we focus on reducing running time, which we denote $R(p) \in \mathbb{R}$. Since we want the fastest correct program, we let $R(p) = -\infty$ if $p$ does not pass one of the given test cases. In practice, measuring a speedup can be difficult due to the stochastic nature of program execution. Recent work has proposed benchmarks that seek to mitigate this issue. The approach used by the PIE benchmark is to measure performance using a system simulator (specifically, gem5 (Binkert et al., 2011)), which

provides deterministic emulation of hardware, enabling fully reproducible results. Finally, we also set $R(p) = -\infty$ if evaluating $p$ in gem5 times out. For Mercury, we set $R(p)$ to be a modified form of Beyond@1, their runtime percentile metric (described in Section 4.1).

To aid adaptation, we assume given a training set of slow-fast program pairs $\Pi = \{(p, p')\}_{j=1}^n$, where $p$ is an unoptimized program and $p'$ is a hand-optimized program; e.g., the PIE benchmark constructs such a dataset based on sequences of submissions from individual participants in competitive programming challenges (Shypula et al., 2024). Given a sequence of submissions $p_1, ..., p_k$, they include pairs $(p_i, p_{i'})$ where $i < i'$ and where $p_{i'}$ is at least 10% faster than $p_i$ according to gem5, i.e., $R(p_{i'}) \geq 1.1 \cdot R(p_i)$. They also provide a subset of *high-quality* training pairs that achieve a more substantial speedup by selecting a subset of the pairs $(p_{i'}, p_i)$ with the highest speedups $R(p_{i'})/R(p_i)$. Using their approach, we also construct a training set for Mercury by selecting high-quality pairs.

Finally, we are interested in blackbox adaptation techniques, which do not adjust the weights of the LLM; instead, they focus on prompting the LLM to improve performance. These prompts can be dynamic (e.g., include dynamically retrieved training examples), multi-turn (e.g., iteratively refine an example based on feedback), or incorporate search (e.g., incrementally apply a sequence of prompts).

**General framework.** We describe the general Retrieval-Augmented Search (RAS) framework for program optimization. RAS assumes that it is given a training set $\Pi_{\text{train}} = \{(p, p')\}_{j=1}^n$ of slow-fast program pairs, and a new program $p_0 \in \mathcal{P}$ to be optimized. In addition, it assumes it is given a retrieval strategy, which can be expressed as a distance function $d : \mathcal{P} \times \mathcal{P} \to \mathbb{R}_{\geq 0}$ between pairs of programs. Typically, the strategy is defined by an embedding model $\phi : \mathcal{P} \to \mathbb{R}^{\bar{d}}$, in which case we can define the distance based on the $L_2$ distance between the embedding vectors of two programs:

$$d_\phi(p, q) = \|\phi(p) - \phi(q)\|$$

Our framework also assumes blackbox access to an LLM $F_{\text{opt}}$, which takes as input an in-context example of a slow-fast program pair $\pi \in \mathcal{P}^2$, along with a new program $p$. Then, we can sample optimized versions $p' \sim F_{\text{opt}}(\pi, p)$ of $p$ from $F_{\text{opt}}$. In our implementation, $F_{\text{opt}}$ is provided with a system prompt instructing it to try and optimize $p$.

Now, RAS performs a variation of beam search to optimize $p_0$, where at each step, it additionally retrieves in-context examples from the training set $\Pi_{\text{train}}$. In particular, at the $i$th iteration of beam search (starting from $i = 1$), let $p_{i-1}$ be the current program. Then, we retrieve the top $k$ programs from $\Pi_{\text{train}}$ to form the in-context dataset:

$$\Pi_i = \text{top-}k\{((p, p'), d(p_{i-1}, p)) \mid (p, p') \in \Pi_{\text{train}}\}.$$

Here, top-$k$ selects the $k$ *new* slow-fast pairs $(p, p')$ with the smallest distances $d(p_{i-1}, p)$, using FAISS (Douze et al., 2024) for vector search. For any retrieved example $\pi_i^j$, we call $\pi_i^j$ a new pair if $F_{\text{opt}}$ did not use $\pi_i^j$ to sample an earlier best-performing program $p_{\text{opt}} \in \{p_1, \ldots, p_{i-1}\}$. Note that retrieval is performed based on the slow program $p$; intuitively, we want a slow program that is similar to $p_{i-1}$ so we can apply similar optimizations to $p_{i-1}$ as the ones encoded by the pair $(p, p')$. Now, for each retrieved example $\pi_i^j \in \Pi_i$, we sample an optimized version of $p_{i-1}$ using $\pi_i^j$:

$$p_i^j \sim F_{\text{opt}}(\pi_i^j, p_{i-1}).$$

Finally, we choose $p_i$ to be the fastest program that correctly passes all test cases:

$$p_i = \underset{j \in [k]}{\arg\max}\, R(p_i^j),$$

where $[k] = \{1, ..., k\}$. If no program passes all of the test cases (i.e., $R(p_i^j) = -\infty$ for all $j \in [k]$), or if all programs time out, then we set $p_i = p_{i-1}$. We continue this process for $m$ steps, producing a sequence of programs $p_1, ..., p_m$. Finally, we return $p_m$. If there is no program at step $m$ that passes all of the test cases and does not time out, we return the source program $p_0$. Note that the hyperparameters of our algorithm are the number of in-context examples $k$ and the number of iterations $m$; we describe the choices we use in our experiments in Section 4.

**Contextual retrieval.** Our instantiation of RAS uses contextual retrieval to identify relevant in-context examples. We compute $\phi(p)$ by first using an LLM $F_{\text{context}}$ to generate a natural language description (i.e., the "context" in contextual retrieval) of $p$ (denoted $s = F_{\text{context}}(p)$), and then applying

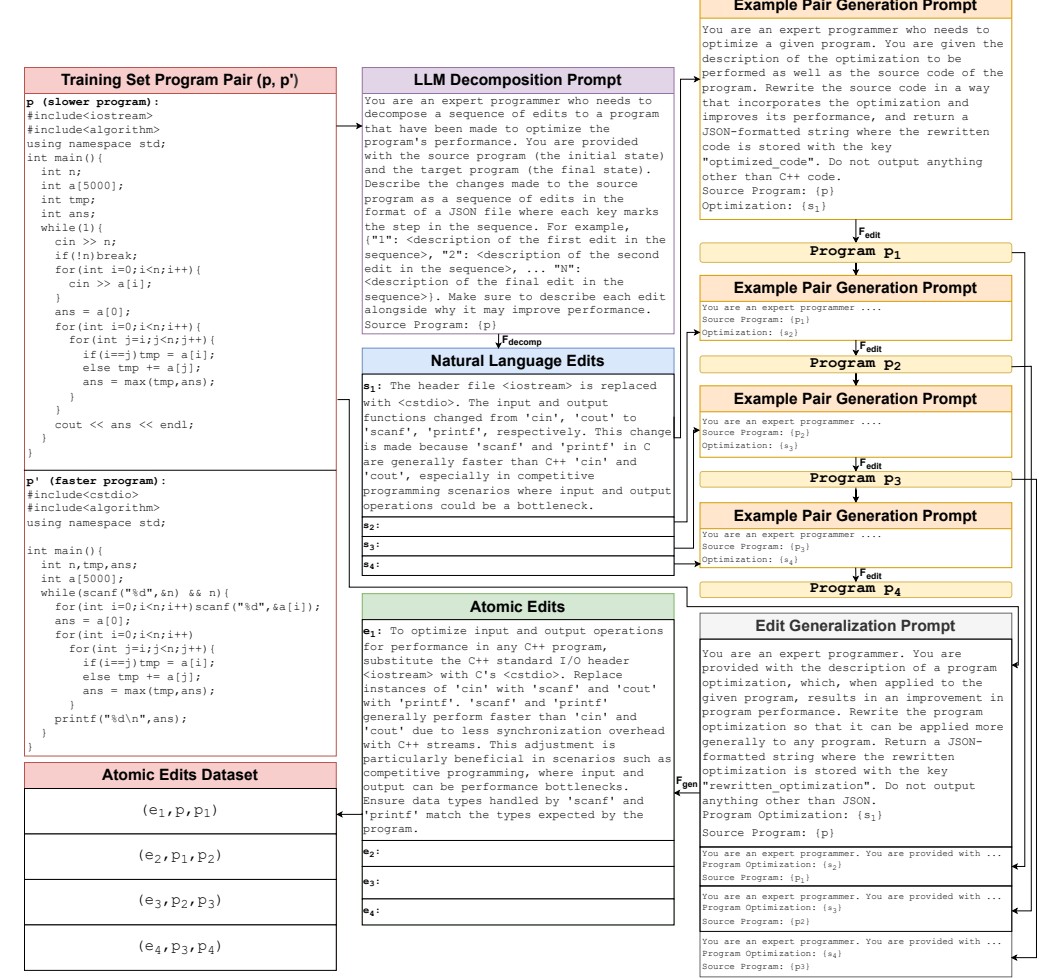

Figure 2: **AEGIS Framework**: For a given training set program pair $(p, p')$, we identify the natural language edits using $F_{\text{decomp}}$, and then generate intermediate programs implementing each edit by using $F_{\text{edit}}$. Finally, the natural language edits are generalized by $F_{\text{gen}}$ to construct atomic edits.

an embedding model $\psi$ to obtain a vector $\psi(s) \in \mathbb{R}^d$, i.e., $\phi(p) = \psi(F_{\text{context}}(p))$. For examples $(p, p') \in \Pi_{\text{train}}$, we can precompute the embeddings, so the LLM $F_{\text{context}}$ only needs to be run once for each one. To construct $F_{\text{context}}$, we use a blackbox LLM that is instructed to describe features like the algorithms and data structures used by the program; this prompt is shown in Figure 1 with an example of a pair $(p, s)$ of program $p$ and its description $s$. Finally, we also compare to an ablation used in prior work on program optimization (Shypula et al., 2024). Here, we directly embed the given program—i.e., $\phi(p) = \psi(p)$ for some embedding model $\psi$; we call this approach *code retrieval*.

## 3  ATOMIC EDIT GUIDED SEARCH

Next, we describe *Atomic Edit GuIded Search (*AEGIS*)*, which is designed to improve interpretability of RAS (overview in Figure 2 and pseudocode in Algorithm 2). AEGIS is inspired by modern compilers, which are designed to perform a sequence of *passes*, which incrementally transform the program to improve performance. Breaking down optimizations into smaller steps has the potential to improve interpretability since the changes from one step to the next may be easier for the programmer to understand. We propose to generate *atomic edits*, which comprise pairs of programs $(p, p')$ that are assumed to be semantically equivalent and roughly differ by a single code optimization.

---

**Algorithm 2** Atomic Edit-Guided Search (AEGIS)

---

**input:** $\Pi_{\text{train}}, F_{\text{decomp}}, F_{\text{edit}}, F_{\text{gen}}, F_{\text{opt}}, F_{\text{context}}, R$
$\Pi_{\text{atomic}} \leftarrow \varnothing$
**for** $(p, p') \in \Pi_{\text{train}}$ **do**
    $[s_1, \ldots s_r] \sim F_{\text{decomp}}(p, p')$
    **for** $i \in [1, ..., n]$ **do**
        $p_i \sim F_{\text{edit}}(s_i, p_{i-1})$
        $e_i \sim F_{\text{gen}}(s_i, p_i)$
        $\Pi_{\text{atomic}} \leftarrow \Pi_{\text{atomic}} \cup \{(e_i, (p_{i-1}, p_i))\}$
**return** $\Pi_{\text{atomic}}$

---

To realize this goal, AEGIS replaces the original training dataset $\Pi_{\text{train}}$ with a dataset of atomic edits $\Pi_{\text{atomic}}$, and then uses RAS in conjunction with $\Pi_{\text{atomic}}$. By retrieving atomic edits, we can guide the underlying LLM $F_{\text{opt}}$ to perform incremental optimizations rather than large changes. AEGIS constructs $\Pi_{\text{atomic}}$ by using an LLM $F_{\text{decomp}}$ to decompose each pair $(p, p') \in \Pi_{\text{train}}$ into atomic edits; then, it aggregates together all discovered atomic edits into the new training set $\Pi_{\text{atomic}}$.

Specifically, we instruct $F_{\text{decomp}}$ to describe the differences between the each slow-fast program pair $(p, p') \in \Pi_{\text{train}}$ as a list; then, the output of $F_{\text{decomp}}$ is a list of *natural language edits* $[s_1, \ldots, s_r] \sim F_{\text{decomp}}(p, p')$, where each $s_i$ is a natural language description of an edit in $(p, p')$. Next, we apply each edit in sequence to the slow program $p$ to obtain a sequence of programs. We do so by initializing $p_0 = p$, and then prompting an LLM $F_{\text{edit}}$ to apply natural language edit $s_i$ to $p_{i-1}$ to obtain the next program $p_i \sim F_{\text{edit}}(p_{i-1}, s_i)$ in the sequence; here, $F_{\text{edit}}$ is instructed to apply the edit to the given program. Assuming the natural language edits accurately describe how $p'$ is obtained from $p$, then the final program $p_r$ in this sequence should resemble the original optimization $p'$ of $p$; in particular, $p_r$ should also be an optimized version of $p$.

We construct our atomic edit dataset using pairs from the resulting sequence. For each tuple $(s_i, p_{i-1}, p_i)$, we ask an LLM $F_{\text{gen}}$ to generalize $s_i$ so it applies to a wider variety of programs; the resulting description $e_i \sim F_{\text{gen}}(s_i, p_i)$ is an atomic edit. Then, our dataset of atomic edits is

$$\Pi_{\text{atomic}} = \bigcup_{(p, p') \in \Pi_{\text{train}}} \{(e_i, (p_{i-1}, p_i))\}.$$

Finally, we can use RAS with $\Pi_{\text{atomic}}$, with a slight modification to account for some of the extra information. Specifically, we modify the LLM $F_{\text{opt}}$ for program optimization to include the atomic edit—i.e., given atomic edit $(e, \pi)$ and program $p$, we sample an optimized version $p' \sim F_{\text{opt}}(e, \pi, p)$. Intuitively, $e$ provides instructions on how to optimize $p$, and $\pi$ shows one example applying $e$.

## 4 EXPERIMENTS

### 4.1 EXPERIMENTAL SETUP

**Benchmark.** Our experiments are based on the PIE benchmark (Shypula et al., 2024), a dataset of slow-fast C++ program pairs constructed from submissions by human programmers to CodeNet (Puri et al., 2021). Since competitive programmers iteratively refine their code submissions for better performance, the authors of PIE construct this dataset by first identifying a sequence of programs submitted by the same programmer to solve a single problem. They filter out any incorrect submissions, and then construct slow-fast pairs by executing the C++ submissions on the gem5 simulator (Binkert et al., 2011) to measure the running time of the code, discarding any pairs whose difference in performance improvement is less than 10%. We use 4080 high-quality pairs from the PIE dataset as our training set $\Pi_{\text{train}}$, and 973 test set pairs as a held-out test set $\Pi_{\text{test}}$. These high-quality pairs are constructed by taking up to 4 pairs in the PIE benchmark's training set with the highest speedup for each competitive programming problem. Importantly, the train-test split in PIE is based on the competitive programming problem being solved, so the training and test set programs are semantically different. For Mercury, we are provided with a training and test set with reference solutions for each problem. We use the same approach on the 1633 Leetcode problems in the training set to construct a high-quality training set $\Pi_{\text{train}}$ of 6372 pairs using Leetcode's reported runtimes for

| Approach | GPT-4o | | Qwen-3-Coder | |
|---|---|---|---|---|
| | **Mean Best Speedup** | **% Optimized** | **Mean Best Speedup** | **% Optimized** |
| RAS | **8.01** | **0.9640** | **8.61** | **0.9774** |
| No Contextual | 5.80 | 0.8520 | 4.52 | 0.6927 |
| Dynamic Retrieval | 4.43 | 0.8191 | 4.23 | 0.7749 |
| Instruct Only | 2.31 | 0.5447 | 1.73 | 0.4018 |
| Human | 3.63 | 0.9887 | 3.63 | 0.9887 |

Table 1: Comparing RAS to baselines on PIE.

| Approach | GPT-4o | |
|---|---|---|
| | **Mean Best Speedup** | **% Optimized** |
| AEGIS | **6.08** | **0.9065** |
| No Contextual | 3.85 | 0.7554 |
| Instruct Only | 2.31 | 0.5447 |
| Human | 3.63 | 0.9887 |

Table 2: Comparing AEGIS to baselines on PIE.

the solutions. We evaluate our approaches on the slowest-provided reference solutions for the 256 held-out problems in Mercury's test set (Du et al., 2024).

**Baselines.** We compare our approach to dynamic retrieval, the highest performing blackbox adaptation strategy studied in PIE (Shypula et al., 2024). This approach also dynamically retrieves in-context examples from $\Pi_{\text{train}}$. There are two key differences between our approach and theirs. First, they use retrieval based on the embedding of the code itself rather than contextual retrieval (i.e., code retrieval). Second, they do not perform sequential search; instead, given a program $p$, they retrieve $k$ in-context examples $\Pi \subseteq \Pi_{\text{train}}$ to provide to the LLM $F'_{\text{opt}}$, and then take multiple samples $p^1, ..., p^h \sim F'_{\text{opt}}(\Pi, p)$. They return the fastest correct program among the $h$ choices.

In addition, we also compare to a "no contextual" ablation of our approach that uses PIE's strategy for retrieval but with search; in particular, it performs code retrieval instead of contextual retrieval. One iteration proceeds as with dynamic retrieval, but we perform multiple iterations. In particular, let $p_0$ be the initial program; on the $i$th iteration (starting from $i = 1$), we sample $k$ in-context examples $\Pi \subseteq \Pi_{\text{train}}$ using code retrieval, draw samples $p_i^1, ..., p_i^h \sim F'_{\text{opt}}(\Pi_i, p_{i-1})$, and then let $p_i = \arg\max_{j \in [h]} R(p_i^j)$; as in RAS, we let $p_i^j = p_{i-1}^j$ if $R(p_i^j) = -\infty$ for all $j \in [h]$.

We also consider a "instruct only" approach from PIE that performs neither retrieval (i.e., does not use $\Pi_{\text{train}}$) nor search; instead, we simply instruct the LLM $F''_{\text{opt}}$ to optimize the given program $p$ to obtain an optimized version $p' = F''_{\text{opt}}(p)$, i.e., $F''_{\text{opt}}$ is an unadapted LLM. The prompt used in the "instruct only" setting is described in Appendix A, and the remaining prompts are described in Appendix B. Finally, we include the "human" speedup—for an initial program $p$, it is the speedup achieved by the fastest correct program $p'$ written by the human participant who wrote $p$. For Mercury, we evaluate on our strongest-baseline, No Contextual, and provide our Instruct Only results for reference.

**Hyperparameters.** For PIE, in our approaches (RAS and AEGIS with contextual retrieval), we use $k = 8$ retrievals and $m = 4$ beam search steps and take $h = 1$ sample per generated prompt. For our baselines, we normalize computation according to the number of calls to the LLM $F_{\text{opt}}$, $F'_{\text{opt}}$, or $F''_{\text{opt}}$. In this calculation, note that for $F'_{\text{opt}}$, the number of retrievals $k = |\Pi|$ does not affect the number of calls $F'_{\text{opt}}(\Pi, p)$, since all examples are included in a single call. Then, for our dynamic retrieval baseline, we retrieve $k = 4$ examples (the same as used in PIE) and take $h = 32$ samples. For our "no contextual" ablation, we retrieve $k = 4$ examples, take $h = 8$ samples per iteration, and use $m = 4$ iterations (the same as our approach). For our "instruct only" ablation, we take $h = 32$ samples and use $m = 1$ iterations. We note that this is different from the standard *pass@k* metric used to evaluate LLM code performance in previous work (such as (Chen et al., 2021), where $k$ refers to the number of samples taken from the LLM, which we denote as $h$ in our case. We use $k$ to denote the number of retrieved examples used in the prompt, as done in (Shypula et al., 2024). For Mercury, we only execute $m = 2$ iterations of RAS (with $k = 8, h = 1$) and no contextual (with $k = 4, h = 8$) and , so we set $h = 16$ for the Instruct Only approach.

| Method | Beyond@1 | Pass@1 |
|---|---|---|
| RAS (Qwen2.5-7B) | **87.85** | **98.83** |
| No Contextual (Qwen2.5-7B) | 69.26 | 97.27 |
| Instruct Only (Qwen2.5-7B) ($h = 16$) | 77.58 | 98.44 |
| Base | 58.66 | 96.88 |

Table 3: RAS Experiments on Mercury. The base values represent the unoptimized test set programs.

**Compute.** For all experiments, we use OpenAI's `gpt-4o-2024-08-06` as $F_{\text{context}}$ for the training set, as well as $F_{\text{decomp}}$, $F_{\text{edit}}$, and $F_{\text{gen}}$ for the PIE atomic edit dataset. We then use the model specified in the experiment as $F_{\text{opt}}$, $F'_{\text{opt}}$, $F''_{\text{opt}}$, and $F_{\text{context}}$ while executing the search procedure. We use OpenAI's `text-embedding-3-large` as the embedding model $\psi$. We run the gem5 simulator on a server with $2\times$ Intel(R) Xeon(R) Gold 6342 CPUs (96 cores total). All C++ programs evaluated in our experiments are compiled using a g++ compiler with the -O3 flag. We use an AWS t2.2xlarge instance for measuring runtime percentiles for Mercury.

**Metrics.** Running gem5 on all test cases to evaluate a single program can be prohibitively computationally expensive due to the large overhead of running gem5. Instead, we measure running time averaged across a subset of 5 randomly selected test cases; these 5 test cases are fixed ahead-of-time. To validate this strategy, we check the correlation between running times on the full test suite vs. our 5 random test cases across all programs in the PIE test set; we find a strong correlation (Pearson's $r = 0.89$, $p < 0.001$; Spearman's $\rho = 0.86$, $p < 0.001$), suggesting that 5 test cases suffices to obtain an accurate estimate of running time. We report results on the held-out test set $\Pi_{\text{test}} \subseteq \mathcal{P}$ of 973 unoptimized programs provided by the PIE benchmark. Our main metric is "mean best speedup"

$$\text{Speedup}(p, p') = \max \left\{ \frac{\text{RunningTime}(p')}{\text{RunningTime}(p)}, 1 \right\}$$

of the final program $p'$ compared to the original program $p$, averaged across all test programs $p \in \Pi_{\text{test}}$, where The minimum speedup is set to 1 since we can always return $p$. We also report "% optimized", which is the number of test programs $p$ for which the optimized program $p'$ is at least $1.1\times$ as fast as $p$. While this metric is not the main goal of our system, it helps capture the diversity of programs that can be optimized using a given approach. For Mercury, we report their Pass@1 and Beyond@1 (mean runtime percentile) metrics (Du et al., 2024), with the modification that fastest generated program using each approach is assumed to be the 1 sample generated by the LLM.

## 4.2 Results

We show C++ results for RAS in Table 1 and for AEGIS in Table 2, and show Python optimization results for RAS in Table 3. First, note that RAS significantly improves performance compared to all baselines, when using both the original PIE training set as well as our atomic edit training set. Dynamic retrieval was by far the best blackbox adaptation approach studied in the original PIE paper, yet our approach is able to almost double its performance in terms of mean best speedup. Our ablation demonstrates that both search and contextual retrieval are roughly equally important, since ablating contextual retrieval about halves the performance improvement compared to dynamic retrieval. While AEGIS diminishes performance, it still achieves a significant improvement. Indeed, it outperforms all ablations (both ablations of AEGIS and those of RAS); the only approach it does not outperform is the full RAS approach. An analysis of the types of programming problems that RAS and AEGIS fail to optimize in our GPT-4o experiments is presented in Appendix D. For Python optimization, we observe that RAS increases the mean runtime percentile metric (Beyond@1) by 10.27 for Qwen2.5-7B, significantly narrowing the gap between it and the Instruct Only performance of larger models. Pass@1 and Beyond@1 results for larger models in the Instruct Only setting are provided in Appendix F.

**Metrics across beam search iterations.** Next, in Figure 3, we study the effect of using search techniques by reporting our various metrics across iterations of beam search on our GPT-4o C++ experiments. We focus on our results for our approach compared to our "No Contextual" ablation (since "Dynamic Retrieval" and "Instruct Only" do not perform search). Figure 3 (a) shows results for "Mean Best Speedup". As can be seen, while the first step of beam search provides the greatest benefit, it continues to provide benefit for all approaches, especially when using contextual retrieval. Since we request the LLM $F_{\text{context}}$ to describe the algorithm used for the current best-performing program

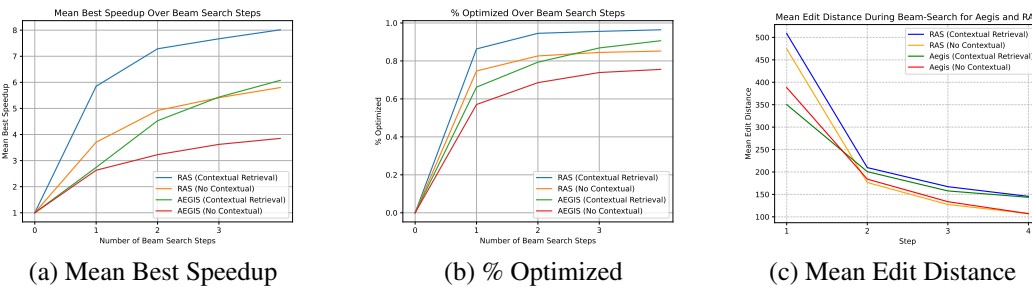

(a) Mean Best Speedup            (b) % Optimized            (c) Mean Edit Distance

Figure 3: Mean Best Speedup, %Optimized, and Mean Edit Distance across beam search steps on GPT-4o experiments.

$p_i$ at each iteration $i$ of the beam search, we hypothesize that $F_{\text{context}}$ can update its description to include algorithmic updates made in the previous iteration, thus enabling it to retrieve more relevant examples. We also see greater continuing improvements for AEGIS, likely because atomic edits constrain optimization to change the program more slowly. Additional iterations may help further close the gap between AEGIS and RAS. We provide an example of how AEGIS and RAS both optimize the same program in Appendix C. Next, Figure 3 (b) shows results for "% Optimized". These results converge substantially more quickly, likely because the first iteration is already enough to get above 1.1× speedup for most programs. Nevertheless, we continue to see gains for our AEGIS approach, again suggesting that continuing search may close the performance gap.

**Accuracy.** RAS and AEGIS are designed to generate programs that pass all test cases; however, this strategy does not ensure correctness. To quantify the error rate of RAS and AEGIS more rigorously on PIE in our GPT-4o experiments, we examined if the programs selected at each step of the procedure would differ if at each iteration of search, we selected the fastest program while ignoring correctness. Across four iterations of search, while selecting for fastest program without measuring accuracy, for RAS, 5/973 test set instances choose an incorrect program (so accuracy is 99.5%), while for AEGIS, 0/973 test set instances chose an incorrect program (so accuracy is 100%). These results suggest that the LLM is highly accurate at producing optimizations that preserve semantic equivalence. We note that LLM-based program optimization systems are already deployed in practice (Shypula et al., 2025), leaving it up to the programmer to validate correctness of the optimizations.

**Interpretability.** A key motivation for AEGIS is that it should provide greater interpretability by making smaller edits. To study this objective, we consider two metrics. Our main metric is the character-level edit distance of pairs of programs $(p_i, p_{i+1})$ encountered as part of the search process, with lower edit distances indicating more incremental changes; we consider the edit distance averaged across all pairs of programs and across all programs in the test set. We summarize results for AEGIS and RAS in Table 5, including results for the "no context" ablations of each approach. As can be seen, AEGIS significantly reduces mean edit distance in both cases. Furthermore, in Figure 3, we show how the mean edit distance changes across steps. As can be seen, AEGIS significantly reduces mean edit distance in the first step, from about 500 to 350. These results suggest that RAS is performing significant optimizations in the first step, and the subsequent steps have smaller edit distance simply because the optimizations are more incremental. Even a single uninterpretable step can make the entire sequence less interpretable, so these results further emphasize the effectiveness of our approach.

## 5  CONCLUSION

We have proposed RAS and AEGIS, two methods for LLM-guided program optimization that incorporate beam search and retrieval to iteratively optimize a given program. We achieve significant speedups in the blackbox setting (i.e., without any fine-tuning), outperforming existing LLM-based program optimization techniques. AEGIS also aims to improve interpretability by decomposing training examples into "atomic edits" that represent incremental optimizations rather than large changes. We believe that our approach provides a compelling strategy for adapting LLMs to code optimization in the blackbox setting, and may be effective in other code generation tasks as well.

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

# A    COMPARING INSTRUCTION PROMPTING AND EXPERT PROGRAMMER SYSTEM ROLES

In our "Instruct Only" baseline, we experiment with two prompts: an instruction-prompting approach (as described in the results of the original PIE benchmark (Shypula et al., 2024)), and an "expert programmer" system role. We provide the exact prompts for our approaches here and whenever we refer to programs or retrieved natural language optimizations, we enclose them in braces. Our prompts are as follows:

## A.1    INSTRUCTION PROMPTING (IP)

Given the program below, improve its performance:

### Program: {Program to be optimized}

### Optimized Version:

## A.2    EXPERT PROGRAMMER SYSTEM ROLE (EPSR)

**System Role**: You are an expert programmer who needs to optimize a given program. You are given the source code of the program. Rewrite the source code in a way that optimizes performance such that the program executes faster, and return a JSON-formatted string where the rewritten code is stored with the key "optimized_code". Do not output anything other than C++ code.
**User Role**: Source Code: {Program to be optimized}

## A.3    PROMPT RESULT COMPARISON

We evaluate the two prompts on our dataset of 973 programs by taking $k = 32$ samples for $m = 1$ iteration of search. Our results are presented in Table 4.

| Approach | Mean Best Speedup | % Optimized |
|----------|-------------------|-------------|
| EPSR | **2.31** | 0.5447 |
| IP | 2.16 | **0.5632** |

Table 4: Results comparing differences in metrics due to prompts in Instruct Only setting

Since we observe a slight increase in Mean Best Speedup in the setting with an expert-level system role, we use it in all our other prompts for to maximize efficacy. The "Instruct Only" setting results we report in Tables 1 & 2 use this expert-programmer system role prompt, which is used by $F''_{\text{opt}}$.
.

# B    PROMPTS FOR EXPERIMENTAL RESULTS

We present our prompts for our PIE experiments below. For our Mercury experiments, we replace all instances of the phrase "C++" with "Python".

## B.1 RAS

### B.1.1 PROGRAM DESCRIPTION GENERATION

This prompt is used by $F_{\text{context}}$.

**System Role**: You are an expert programmer who has been provided with a program solving a programming problem, called the source program. You need to identify the algorithm being used to solve the problem, and your goal is to generate a JSON object with the key "algorithm" which has the value as one sentence describing the algorithm used in the code snippet.
**User Role**: Source Program:
{Program to be optimized}

### B.1.2 GENERATING PROGRAMS WITH CONTEXTUAL RETRIEVAL

This prompt is used by $F_{\text{opt}}$.

**System Role**: You are an expert programmer who needs to optimize a given program, called the source program. You are given one pair of fast and slow programs as an example, which are presented as a pair where "slower version" refers to the slow code and "optimized version" refers to the faster, more optimal version of the same program. The last program with the label "slower version" is the source program whose optimized version you need to generate. Rewrite the source program in a way that incorporates all of the optimizations in the example, and return a JSON-formatted string where the rewritten code is stored with the key "optimized_code". Do not output anything other than C++ code.
**User Role**:
# slower version:
{Retrieved Slow Program}
# optimized version of the same code:
{Retrieved Faster Program}

# slower version:
{Program to be optimized}
# optimized version of the same code: \n

### B.1.3 GENERATING PROGRAMS WITH DYNAMIC CODE RETRIEVAL

This is the prompt used in both the "No Contextual" and "Dynamic Retrieval" settings for RAS, as well as the "No Contextual" setting for AEGIS. It is passed to the model $F'_{\text{opt}}$.

---

**System Role**: You are an expert programmer who needs to optimize a given program, called the source program. You are given several pairs of fast and slow programs, called examples, which are presented as pairs where "slower version" refers to the slow code and "optimized version" refers to the faster, more optimal version of the same program. The very last program with the label "slower version" is the source program whose optimized version you need to generate. Rewrite the source program in a way that incorporates all of the optimizations in the examples, and return a JSON-formatted string where the rewritten code is stored with the key "optimized_code". Do not output anything other than C++ code.

**User Role**:
# slower version:
{Retrieved Slow Program 1}
# optimized version of the same code:
{Retrieved Faster Program 1}

# slower version:
{Retrieved Slow Program 2}
# optimized version of the same code:
{Retrieved Faster Program 2}

# slower version:
{Retrieved Slow Program 3}
# optimized version of the same code:
{Retrieved Faster Program 3}

# slower version:
{Retrieved Slow Program 4}
# optimized version of the same code:
{Retrieved Faster Program 4}

# slower version:
{Program to be optimized}
# optimized version of the same code: \n

---

## B.2 AEGIS

### B.2.1 GENERATING NATURAL LANGUAGE EDITS

This prompt is used by $F_{\text{decomp}}$.

> **System Role**: You are an expert programmer who needs to decompose a sequence of edits to a program that have been made to optimize the program's performance. You are provided with the source program (the initial state) and the target program (the final state). Describe the changes made to the source program as a sequence of edits in the format of a JSON file where each key marks the step in the sequence. For example, "1": <description of the first edit in the sequence>, "2": <description of the second edit in the sequence>, ... "N": <description of the final edit in the sequence>. Make sure to describe each edit alongside why it may improve performance.
> **User Role**:
> Source Program: {Slow Program from Training Set Program Pair}
> Target Program: {Faster Program from Training Set Program Pair}

### B.2.2 GENERATING PROGRAM SEQUENCE FROM NATURAL LANGUAGE EDITS

This prompt is used by $F_{\text{edit}}$.

> **System Role**: You are an expert programmer who needs to optimize a given program. You are given the description of the optimization to be performed as well as the source code of the program. Rewrite the source code in a way that incorporates the optimization and improves its performance, and return a JSON-formatted string where the rewritten code is stored with the key "optimized_code". Do not output anything other than C++ code.
> **User Role**:
> Source Program: {Previous Program in Sequence}
> Optimization: {Optimization to be applied to generate next program in the sequence}

### B.2.3 GENERATING ATOMIC EDITS FROM NATURAL LANGUAGE EDITS

This prompt is used by $F_{\text{gen}}$.

> **System Role**: You are an expert programmer. You are provided with the description of a program optimization, which, when applied to the given program, results in an improvement in program performance. Rewrite the program optimization so that it can be applied more generally to any program. Return a JSON-formatted string where the rewritten optimization is stored with the key "rewritten_optimization". Do not output anything other than JSON.
> **User Role**:
> Program Optimization: {Natural Language Edit}
> Program: {Program in program sequence that the edit was applied to}

### B.2.4 GENERATING PROGRAMS WITH CONTEXTUAL RETRIEVAL

This prompt is used by the modified $F_{\text{opt}}$ when generating programs with AEGIS.

> **System Role**: You are an expert programmer who needs to optimize a given program, called the source program. You are given the description of an optimization that is to be performed on the given program, as well as an example showing how to apply the optimization on an example program (called the example source) to get a target program (called the example target). Rewrite the source code in a way that incorporates all of the optimizations, and return a JSON-formatted string where the rewritten code is stored with the key "optimized_code". Do not output anything other than C++ code.
> **User Role**: Source Program:
> {Program to be optimized}
> Optimization:
> {Atomic edit retrieved via contextual retrieval}
>
> Example Source:
> {Slower program in retrieved example pair}
> Example Target:
> {Faster program in retrieved example pair}

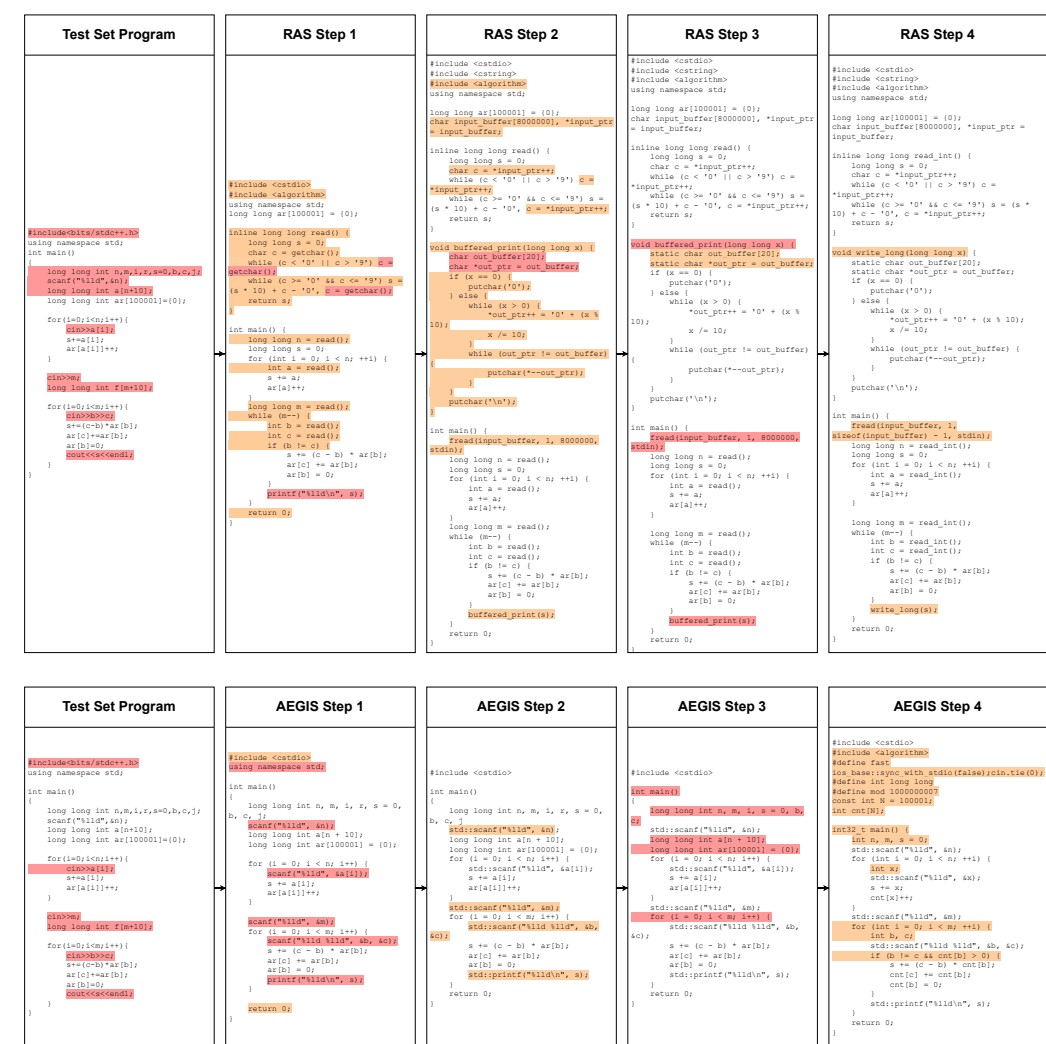

Figure 4: We show a randomly selected example optimization trajectory where RAS and AEGIS implement similar optimizations to achieve similar speedups. The final speedup of RAS is $10.06\times$, compared to $9.58\times$ for AEGIS. We have highlighted lines that have changed from the previous step in orange, while lines that change in the next step have been highlighted in red. For reference, the human speedup on this example is $1.8\times$. Here RAS implements an optimization to replace `cin` and `cout` alongside an optimization to ensure that variables are not needlessly updated when `b = c` in Step 1. AEGIS implements a `cin` and `cout` replacement in Step 1 and refines it until Step 3, and then implements the `b!=c` check in Step 4.

## C  COMPARISON BETWEEN RAS AND AEGIS

In Figures 4 and 5, we show an example of the optimization trajectory taken by each RAS and AEGIS in our GPT-4o experiment on the PIE dataset. As can be seen, RAS concentrates a large number of edits in the first step. In contrast, the edits performed by AEGIS are spread out more evenly across different steps.

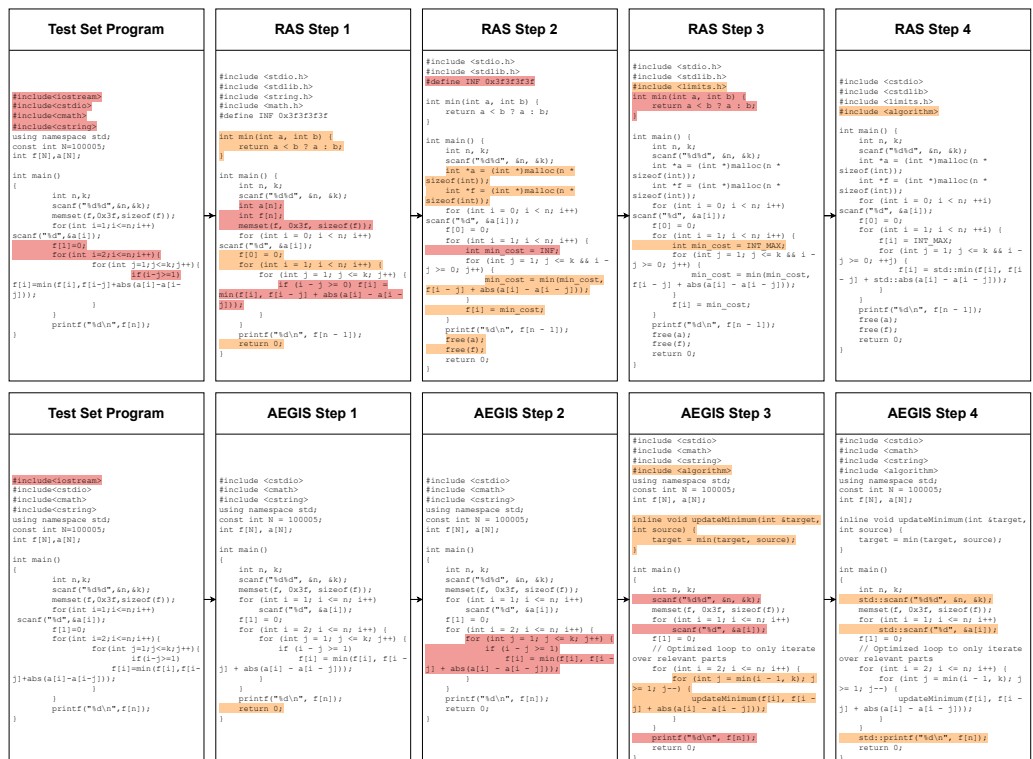

Figure 5: We show a randomly selected example optimization trajectory where RAS significantly outperforms AEGIS. Here, we demonstrate the improvements made at each step of RAS vs. AEGIS. The final speedup of RAS on this example is 7.34×, compared to 2.35× for AEGIS. We have highlighted lines that have changed from the previous step in orange, while lines that change in the next step have been highlighted in red. For reference, the human speedup on this example is 1.37×.

# D    FAILURE CATEGORY ANALYSIS OF RAS AND AEGIS

In our GPT-4o experiment on the PIE dataset, for each of our two methods, AEGIS and RAS, we construct a set of unoptimized programs. The set contains test set programs for whom the final best speedup after using the method is less than 1.1×. We then study the program descriptions generated by the LLM $F_{\text{context}}$ during contextual retrieval. By examining the LLM-generated program descriptions of the test set programs in the unoptimized program set, we can measure the frequency of specific frequently occurring terms. We then compare the frequency of these terms in the unoptimized program set to the overall test set to measure if each method fails disproportionately on problems involving certain algorithms or data structures.

AEGIS appears to struggle with optimizing two primary groups of programs: programs that include dynamic programming algorithms and programs that involve binary search over either sorted lists or trees. Across the complete test set, programs with descriptions including the term "dynamic programming" constitute 51.59% of programs and those mentioning "binary search" constitute 4.32 of programs. AEGIS's set of unoptimized programs constitutes 10.48% of the entire test set. Programs with descriptions including the term "dynamic programming" constitute 48.04% and those with descriptions mentioning "binary search" constitute 10.78% of this unoptimized set. These results suggest that AEGIS fails disproportionately on binary search problems.

For RAS, the unoptimized program set is 3.91% of the entire test sets. When compared to AEGIS, the percentage of dynamic programming problems in RAS's unoptimized program set decreases to 21.05%, while the percentage of binary search problems remains at a similar level at 10.53%. Additionally, for RAS, we observe a high failure rate for programs whose descriptions mention Kruskal's algorithm: while such problems constitute 1.03% of the total test set, they constitute

15.79% of RAS's unoptimized program set. Only 2.94% of problems in AEGIS's unoptimized program set mention Kruskal's algorithm, and we observe that RAS fails on a greater number of problems involving Kruskal's algorithm as compared to AEGIS.

Thus, we can develop greater insights into their failure modes by analyzing the artifacts produced during RAS and AEGIS. By identifying algorithms and data structures that cause these failures, we can subsequently augment the training dataset which we use for retrieval by targeting specific categories of optimizations for subsequent improvements in performance.

## E    MEAN EDIT DISTANCE

We show mean edit distances in Table 5.

| Method | GPT-4o Mean Edit Distance |
|---|---|
| AEGIS | **213.05** |
| RAS | 257.77 |
| AEGIS (No Contextual) | **203.24** |
| RAS (No Contextual) | 221.49 |

Table 5: Comparisons of edit Distances over Steps between AEGIS and RAS on the PIE Benchmark.

## F    MERCURY RESULTS FOR LARGER MODELS

We show results of using larger models to optimize programs in the Mercury dataset in the Instruct Only setting in Table 6.

| Method | Beyond@1 | Pass@1 |
|---|---|---|
| Instruct Only (Gemma3-12B) ($h = 16$) | 88.76 | 99.61 |
| Instruct Only (Qwen3-30B) ($h = 8$) | 91.02 | 99.22 |
| Instruct Only (Qwen-QwQ-32B) ($h = 8$) | **93.13** | **99.61** |

Table 6: RAS Experiments on Mercury on larger models.

## G    LLM USAGE

LLMs were used to generate the code for executing some of the experiments, which was reviewed by the authors.

