# OpenReview forum: "LLM Program Optimization via Retrieval Augmented Search"
_ICLR.cc/2026/Conference — ICLR 2026 Conference Withdrawn Submission_

### Official Review · Reviewer_EKt9 · 2025-10-25

**Soundness:** 2
**Presentation:** 3
**Contribution:** 2
**Rating:** 4
**Confidence:** 4

**Summary:**

The paper proposes Retrieval-Augmented Search (RAS) and an atomic-edit pipeline (AEGIS) for black-box program optimization with LLMs. RAS iteratively retrieves examples via natural-language program descriptions and performs beam-style search; AEGIS decomposes slow-fast pairs into atomic edits to improve interpretability. Reported results on PIE and Mercury show that the proposed RAS method can improve the efficiency of the generated code.

**Strengths:**

+ Clear problem framing for black-box adaptation to performance optimization.
+ The proposed method is rational, combining contextual retrieval and iterative search to improve the efficiency of the generated code.
+ Experiments show that AEGIS improves edit granularity and interpretability.

**Weaknesses:**

- Comparisons center on closely related prompt-engineering variants (dynamic retrieval, no-contextual retrieval, instruct-only). Missing are stronger and more diverse baselines: e.g., white-box adaptation (e.g., fine-tuning), stronger compiler optimization pipelines, or more recent LLM-based code optimization methods. As a result, the superiority of RAS may mostly reflect an advantage within a narrow family of RAG methods rather than against the broader state of the art.
- The paper does not report wall-clock optimization running time in any of the baseline or proposed method.
- The paper lacks a discussion of correctness after optimization. The paper lacks analysis of the correctness after LLM-based optimization is applied. The reviewer is not sure that if the proposed RAS generates faster but incorrect code solutions?

**Questions:**

- What is the post-optimization correctness rate? Do you observe bugs introduced by optimization, and with what frequency and types?

---

> ### Author Response · Authors · 2025-12-03
> **Response (1/1)**
>
> Thank you for your feedback and for acknowledging that our methods are rational and that AEGIS improves interpretability.
>
> **Regarding Comparisons with Other Optimization Methods**
>
> All our C++ experiments are compiled using the g++ compiler with the -O3 flag, which enables aggressive compiler optimizations of the program. Our methods are targeted for LLM-guided optimization of algorithms and data structures, which are well beyond the scope of existing compiler-based approaches. As a result, we significantly outperform the base compiler-optimized programs.
>
> Other works already cover the comparisons between white-box retrieval-based techniques, such as finetuning (PIE) and reinforcement learning (Afterburner). We believe it is well-established that white-box optimization techniques outperform black-box optimization approaches in program optimization; however, white-box approaches are both financially and computationally expensive. Our results highlight that recent models enable black-box optimization approaches to outperform previously reported state-of-the-art fine-tuning results on PIE on GPT-3.5 (6.86x mean performance on pass@8), surpassing previous state-of-the-art black-box optimization approaches.
>
> **Regarding Correctness**
>
> All our methods are correct by construction with respect to the test cases of the associated competitive programming problem. RAS and AEGIS evaluate the generated programs after each iteration, retaining only those that successfully pass all test cases (lines 204-207). Additionally, we run experiments (lines 455-464) to see whether our results are affected in the event that we simply select the fastest program (as evaluated on five test cases) at each iteration without examining correctness; our results show that RAS has 99.5% accuracy in this setting, while AEGIS has 100% accuracy, suggesting that it is extremely rare for the fastest program generated at any iteration by our approaches to fail any of our test cases.
>
> **Wall-Clock Optimization Time**
>
> We report relative ratios between wall-clock runtimes in terms of the mean speedup - we are happy to commit to including raw wall-clock runtimes in our final code release.
>
> **References**
> 1. Shypula, A. G., Madaan, A., Zeng, Y., Alon, U., Gardner, J. R., Yang, Y., Hashemi, M., Neubig, G., Ranganathan, P., Bastani, O., & Yazdanbakhsh, A. (2024). Learning Performance-Improving Code Edits. In _The Twelfth International Conference on Learning Representations._
> 2. Du, M., Tuan, L. A., Liu, Y., Qing, Y., Huang, D., He, X., Liu, Q., Ma, Z., & Ng, S. K. (2025). Afterburner: Reinforcement learning facilitates self-improving code efficiency optimization. _NeurIPS 2025._

---

### Official Review · Reviewer_SQcB · 2025-10-26

**Soundness:** 2
**Presentation:** 2
**Contribution:** 2
**Rating:** 4
**Confidence:** 3

**Summary:**

This paper presents two black-box adaptation methods for Large Language Models (LLMs) to optimize programs: Retrieval Augmented Search (RAS) and Atomic Edit Guided Search (AEGIS). RAS iteratively optimizes the program through a retrieve-optimize-evaluate loop. At each step, it retrieves the "slow-fast" program pairs from a training set to guide LLM generation using the similarity of the natural language summarization instead of using code retrieval directly. AEGIS aims to improve the interpretability of program optimization by using a sequence of "atomic edits. On PIE and Mercury, RAS significantly outperforms the baselines, and AEGIS can reduce the average edit distance.

**Strengths:**

1. The proposed RAS is simple yet effective.

2. The proposed RAS shows significant improvements on the benchmark, even surpassing human baselines.

**Weaknesses:**

1. Although the proposed method RAS improves the optimization performance, it requires too many LLM calls ($m \times k$), which is costly in practice.

2. The authors prove that contextual retrieval can outperform code retrieval. However, it seems counterintuitive because the raw code should naturally reflect more details than code descriptions. I wonder if it is just because the embedding model used in the paper cannot handle the code input well.

3. RAS in fact is different from beam search as it does not keep top-k but only the best candidates at each step.

4. The value of AEGIS is uncertain. First, it does not reach the same level of optimization as RAS, although it involves more LLM calls to process the dataset. Second, although the paper claims that it can improve the interpretability of the generated samples by showing the decrease of the edit distance, this metric is not straightforward to represent the interpretability as sometimes the optimization requires changing high-level algorithms and data structures (also mentioned in Line 82)

5. The descriptions of Algorithm 1 and Algorithm 2 need to be improved. Some notions are not used in the workflow (e.g. $F_{context}$)

**Questions:**

1. Could you provide more insights on why contextual retrieval can outperform code retrieval?

2. In AGEIS, why do we need to generalize from $s_i$ to $e_i$ (Line 297)? How does it generalize exactly?

---

> ### Author Response · Authors · 2025-12-03
> **Response (1/2)**
>
> Thank you for your feedback and for recognizing that our approach outperforms existing state-of-the-art approaches for program optimization.
>
> **Regarding Computational Expenditure**
>
> Our primary conclusion pertains to the distribution of computation, assuming a fixed number of “optimization” LLM calls, i.e., LLM calls that generate programs. We argue that devoting a larger percentage of computational effort to identifying relevant examples and iteratively optimizing programs helps boost performance. In each of our approaches and baselines, the same number of programs (32 for PIE and 16 for Mercury) is generated. Our argument does not require comparing each approach to prompting a model once; a fairer comparison is to compare sampling programs at pass@32 versus conducting a search in RAS and AEGIS to make $m \times k = 32$ calls.
>
> The only difference between the number of calls made in RAS and in the Instruct-Only setting comes from the LLM calls made for generating program descriptions and embeddings. For the training dataset, these calls are a fixed cost that does not scale with the number of test set programs optimized. At test time, an additional call is required to generate the description (and the corresponding description embedding) of the best-performing program at each iteration of the search. Thus, RAS only needs $m$ more LLM calls than a standard $pass@32$ evaluation, where $m$ is the number of iterations of search performed and can be tuned to the user’s needs. Thus, we do not believe that our approach is significantly more computationally expensive than standard evaluation methods for evaluating LLM-guided program optimization using multiple LLM-generated samples.
>
> **Regarding the Relative Performance of Contextual and Dynamic Retrieval**
>
> We believe that previous research has shown that in different retrieval settings (such as documents), contextual retrieval outperforms embeddings constructed from the retrieval contents. In the context of code snippet retrieval, other previous works, such as CoaCor, have shown that retrieving code snippets via text annotations is a viable retrieval technique. We do not believe that the performance gains can simply be attributed to a failure of the embedding model, since, on inspecting the examples retrieved by both approaches, we find that dynamic retrieval often retrieves examples that have superficial similarity to the test set example, such as shared variable names and overlapping distribution of functions and objects. However, on using contextual retrieval, we are able to surface program pairs that share algorithmic and data structure similarities, e.g., when given a dynamic-programming-based program, contextual retrieval surfaces other examples of optimized dynamic programming solutions.
>
> **Value of AEGIS**
>
> A crucial component of the AEGIS dataset that differentiates it from RAS is that each retrieved pair is associated with a generalized edit aimed at improving the performance. As a result, we believe that by identifying the generalized atomic edit associated with each retrieved example, AEGIS provides additional insight into the strategies that led to the performance gain. In contrast, if a user were to try to gain the same visibility into the retrieved examples selected by RAS, they would have to identify (in natural language) the optimizations implemented in the retrieved pair selected at each stage of search, and then examine which of those optimizations was applied to the test set problem. We believe that while optimizing safety-critical code using LLMs, AEGIS may be the superior approach, as it provides a more comprehensive picture of the optimizations AEGIS was attempting to implement through its retrieved examples.
>
> **Notation**
>
> $F_\text{context}$ is used to construct the text descriptions and embeddings in Figure 1. Could you please elaborate on which notations you would like us to specify in the workflow?
>
> **Generalization in AEGIS**
>
> The purpose of generalization from $s_i$ to $e_i$ in AEGIS is to prevent the language model from referencing specific function names or variable names from the original training set program pair, since that may cause confusion when included in the prompt to the LLM.
>
> For example, here is an example of an optimization $s_i$:
>
> Replaced the array 'judge' of size 1,000,000 with a vector of boolean 'is_prime' that gets its size set dynamically to the maximum input received, 'max_n'. This reduces memory usage when inputs are smaller than 1,000,000, which can improve cache efficiency and overall performance.
>
> And here is a generalized edit $e_i$:
>
> Replace large fixed-size arrays with dynamically-sized data structures like vectors, specifically allocating only as much memory as needed based on the maximum input or requirement. This dynamic allocation helps reduce unnecessary memory usage, enhances cache efficiency, and can improve overall performance.

---

> > ### Author Response · Authors · 2025-12-03
> > **Response (2/2)**
> >
> > **References**
> >
> > 1. Anthropic (2024). Contextual retrieval.
> > 2. Merola, C., & Singh, J. (2025, April). Reconstructing context: Evaluating advanced chunking strategies for retrieval-augmented generation. In International Workshop on Knowledge-Enhanced Information Retrieval (pp. 3-18). Cham: Springer Nature Switzerland.
> > 3. Yao, Z., Peddamail, J. R., & Sun, H. (2019, May). Coacor: Code annotation for code retrieval with reinforcement learning. In The world wide web conference (pp. 2203-2214).

---

### Official Review · Reviewer_BrAL · 2025-10-31

**Soundness:** 2
**Presentation:** 2
**Contribution:** 2
**Rating:** 2
**Confidence:** 3

**Summary:**

This paper proposes Retrieval Augmented Search (RAS) and Atomic Edit Guided Search (AEGIS) for program optimization using large language models. The key idea is to perform iterative beam search where, at each step, training examples are retrieved based on LLM-generated natural language descriptions of the current program rather than direct code similarity. AEGIS additionally decomposes training pairs into sequences of "atomic edits" to enable more incremental modifications. Experiments on the PIE (C++) and Mercury (Python) benchmarks show substantial improvements over prior methods.

**Strengths:**

- The core idea of combining iterative search with contextual retrieval is intuitive and well-motivated.
- The paper demonstrates substantial improvements over baseline methods, achieving 2× better speedup compared to dynamic retrieval on the PIE benchmark, with consistent gains on the Mercury benchmark as well.
- AEGIS's attempt to improve interpretability through atomic edits is an interesting method, even though the performance trade-off is not ideal.

**Weaknesses:**

**1. Writing and presentation issues**

The paper's organization and clarity need improvement. For example:

- In the "Problem formulation" section (line 152), “Problem formulation. In the program optimization problem, the goal is to take a program $p \in P$ as input, and output an optimized program $p′ \in  P$ that is semantically equivalent to p.” spends an entire paragraph discussing semantic equivalence rather than defining the "optimization" (runtime? memory? how is it measured?).
- Line 86 mentions "retrieval and search are not integrated" in SBLLM, which is confusing since retrieval and search are conceptually similar. The same issue applies to the proposed "retrieval augmented search"—this needs clearer explanation in the introduction.
- There are many vague references throughout: "This existing approach" (line 42), "They find..." (line 33), "they propose..." (line 33) without clear antecedents or citations.

**2. Limited baseline comparisons**

The paper only compares against one established baseline (SBLLM). The related work section is brief and lacks a systematic overview of recent progress in LLM-based code optimization. If this domain hasn't seen much recent work, it would be valuable to discuss why—what are the key challenges? Additionally, the paper could strengthen its evaluation by including generic but effective methods like CoT or ReAct, or some generic agent workflow as baselines.

The choice of PIE benchmark is not well justified. I’ve seen some other code optimization benchmarks in recent year [1,2]. A discussion of benchmark selection would be helpful.

[1] SWE-Perf: Can Language Models Optimize Code Performance on Real-World Repositories?

[2] CodeScope: An Execution-based Multilingual Multitask Multidimensional Benchmark for Evaluating LLMs on Code Understanding and Generation

**3. Unfair computational comparison**

The comparison appears unfair: RAS uses k=8 retrievals times m=4 beam search steps (32 total retrievals), while the baseline uses k=4 retrievals × 1 step (4 total). To ensure fair comparison, the total retrieval volume across all methods should be the same.

Additionally, the paper claims to "normalize computation" by counting LLM calls, but this ignores the cost of generating natural language descriptions for train set, and the preprocessing cost for AEGIS. How the LLM calls are controlled is not explicitly or formally clarified either. From my understanding, sampling h candidates from baseline method only need 1 LLM call. While the multistep RAS need m LLM call per sample.

**4. Unclear source of improvement for RAS**

While RAS shows improvements over the baseline, it's unclear whether the gains come from the method itself or simply from using more computation (4 steps). The baseline does one-pass generation with only more sampled candidates, while RAS does multi-step agent-like retrieval and generation. A fairer comparison would control for total computational budget (such as token) or provide ablations showing diminishing returns.

**5. Limited novelty of contextual retrieval**

The core contribution of RAS—retrieving based on natural language descriptions rather than direct code embeddings—has been widely and extensively studied and applied in prior work. The paper should better articulate what is novel beyond applying an existing technique to this setting.

**6. AEGIS shows degraded performance**

Table 2 shows AEGIS (6.08× speedup) actually performs worse than RAS alone (8.01× in Table 1), despite being the paper's second major contribution. The smaller edit distance is interesting but doesn't offset the performance loss. Why should we prefer AEGIS over RAS?

**Questions:**

1. Can you provide more details on how computational costs are controlled across baselines?

---

> ### Author Response · Authors · 2025-12-03
> **Response (1/3)**
>
> Thank you for your feedback, and for recognizing that our approach substantially outperforms prior state-of-the-art approaches.
>
> **Writing Issues**
>
> We commit to addressing all raised writing issues prior to publication.
>
> **Computational Cost Control**
>
> We would like to clarify the following about our baselines: we do not compare against SBLLM. We compare our approach with dynamic retrieval, the method presented in the original PIE paper. The reason behind this is that SBLLM underperforms dynamic retrieval by a significant margin (1.55x as compared to 2.53x for dynamic retrieval using GPT-4), suggesting that dynamic retrieval is the current state-of-the-art black-box optimization method in this setting.
>
> We compare our method (RAS) to three baselines: Instruct-Only (No Retrieval), Dynamic Retrieval, and No Contextual (which uses dynamic retrieval for retrieving examples, but is also equipped with search).
>
> We normalize computation based on the number of optimization calls to the LLM, i.e., $F_{\text{opt}}$, $F_{\text{opt}}’$, and $F_{\text{opt}}’’$. This means that the output of any individual approach is always 32 generated programs. This count does not include any calls to generate text descriptions and contextual embeddings.
>
> The distribution of calls for PIE is as follows (for Mercury, the number of iterations is 2, so the number of calls for all approaches is scaled accordingly to generate 16 programs in total):
>
> 1. For RAS and AEGIS: For each test set program, we first generate an LLM-generated description. Then, using the description’s embedding, we select 8 programs, from which we construct 8 prompts (1 prompt per retrieved program). Then, we prompt the LLM with each prompt to generate 8 programs, which are then evaluated to identify the best generated program. This is repeated for 4 iterations, resulting in 32 generated programs.
> 2. For the “No Contextual” setting, we use the test set program’s embedding to retrieve 4 programs. Using the retrieved programs, we construct a single prompt that is used to prompt the LLM to generate 8 samples. The best-performing program is identified, and this approach is repeated for 4 steps, resulting in 32 generated programs.
> 3. For the “Dynamic Retrieval” setting, we use the test set program’s embedding to retrieve 4 programs. Using the retrieved programs, we construct a single prompt that is used to prompt the LLM to generate 32 samples in a single setting.
> In the “Instruct-Only” setting, we do not retrieve any programs and prompt the LLM to optimize the existing test-set program. We collect 32 samples from the 1 prompt we construct for this purpose.
>
> We use $h$ to denote the number of samples generated by the LLM per constructed prompt, so for PIE, $h=1$ for RAS and AEGIS, $h=8$ for “No Contextual”, and $h=32$ for “Dynamic Retrieval” and “Instruct-Only. Regardless of the value of h, each method generates 32 programs for evaluation.
>
> We respectfully disagree that we unfairly normalize compute. In fact, our experimental design was shaped by some of the conclusions you have highlighted. Our primary conclusion is that by combining search with techniques that surface more relevant examples, we can significantly improve LLM program optimization capabilities. We agree that the superiority of contextual retrieval over dynamic retrieval is well-established in previous literature. Therefore, we do not directly compare contextual retrieval with dynamic retrieval by normalizing the number of retrieved examples. We believe that prior research has provided sufficient evidence that contextual retrieval, in general, will surface more relevant examples if the number of examples remains the same. The second reason why we do not modify the number of retrieved examples is that in the original PIE paper, the performance difference for GPT-4 and GPT-3.5 with K=2 and K=4 retrieved examples is extremely similar (within 0.08x speedup for pass@1 and within 0.05x for pass@8 in Table 9). Additionally, programs in PIE are also quite long, and our overall LLM generation costs would increase if we included several programs in a single prompt to maintain a constant retrieval volume. As a result, we believe that including more examples of slow-fast pairs in the prompt would yield comparable or worse performance, while being more expensive, as it would provide excessive context to the LLM.

---

> > ### Author Response · Authors · 2025-12-03
> > **Response (2/3)**
> >
> > **Computational Cost Control (contd.)**
> >
> > Instead, to support our primary claim that combining search and retrieval techniques for black-box optimization is a more effective method of improving performance, we normalize against the number of programs generated. Generating all programs through a single iteration of dynamic retrieval (the previous state-of-the-art approach) performs poorly. Augmenting dynamic retrieval with search improves performance quite significantly (the “No Contextual” ablation). Finally, expanding the diversity of programs generated by retrieving more relevant examples through contextual retrieval and using only one example per prompt has the most significant impact, substantially improving the performance of generated programs. We respectfully disagree that our experiments do not effectively isolate our sources of improvement, as our conclusions regarding the source of improvement clearly highlight that search alone does not make a significant difference.
> >
> > Finally, we would also like to respectfully disagree that restricting computational budgets based on total tokens would be a better method for isolating the contributions of individual components. Since the length and nature of prompts diverge significantly across our evaluated methods, holding different approaches to the same number of tokens would severely impact performance when there are more examples to reason over, and likely disadvantage our baselines that include multiple examples in a single prompt.
> >
> > **Benchmark Selection**
> >
> > We respectfully disagree that our evaluation has limited practical applications due to our choice of benchmark. Repository-level benchmarks, such as SWE-Perf, require an understanding of the broader code context. However, as a result, improvements in performance can be made in their repository-level “realistic” setting simply by choosing an appropriate agentic framework (for example, we can examine Claude’s performance on SWE-Perf in the oracle setting versus the realistic setting using OpenHands). In SWE-Perf’s oracle setting, where all relevant files are provided to the LLM as context, it is unclear whether LLM failures are due to 1) inability to identify relevant context from the provided files, 2) incorrect understanding of which optimizations are relevant to the problem, 3) incorrect code generation or application of an optimization to the problem. While issues 2 and 3 are often language model-level failures, issue 1 can often be mitigated by designing multi-agent systems equipped with tool use (as can be seen by TRAE’s performance on the SWE-Bench Verified Leaderboard). As a result, we believe it is essential to develop both black-box and post-training methods for mitigating issues 2 and 3, which can later be integrated into agentic systems that address issue 1.
> >
> > Additionally, we believe that there is widespread recognition that issues 2 and 3 are challenging problems that require dedicated benchmarks within the community. There has been significant interest in recent years in evaluating model performance on issues 2 and 3 independently. Benchmarks such as PIE (ICLR 2024), EffiBench (NeurIPS 2024), Mercury (NeurIPS 2024), ENAMEL (ICLR 2025), and Venus (NeurIPS 2025) all study the optimization of single-file programs and have demonstrated that frontier models still struggle to identify and apply program optimizations that successfully speed up such single-file programs. While several benchmarks have been proposed to study this problem effectively, we have not found a comparable number of works that study approaches to improve model performance on these tasks. Thus, a significant gap exists in the literature regarding methods to improve LLM-guided performance optimization on these single-file program benchmarks, and our work aims to address this gap.
> >
> > Finally, we would like to highlight the scope of our work. While SWE-Perf comprises 140 tasks from 9 GitHub repositories, we use benchmarks that both have more instances (973 test-set problems for PIE and 256 for Mercury) and are drawn from a much wider set of sources (CodeNet problems for PIE and LeetCode problems for Mercury). As a result, our methods can evaluate performance for a wide range of algorithmic optimizations and implementations, allowing us to analyze the distribution of failed optimization implementations in Appendix D - tasks that we believe would be challenging in repository-based benchmarks such as SWE-Perf.

---

> ### Author Response · Authors · 2025-12-03
> **Response (3/3)**
>
> **Preference of AEGIS over RAG**
>
> A crucial component of the AEGIS dataset that differentiates it from RAS is that each retrieved pair is associated with a generalized edit aimed at improving the performance. As a result, we believe that by identifying the generalized atomic edit associated with each retrieved example, AEGIS provides additional insight into the strategies that led to the performance gain. In contrast, if a user were to try to gain the same visibility into the retrieved examples selected by RAS, they would have to identify (in natural language) the optimizations implemented in the retrieved pair selected at each stage of search, and then examine which of those optimizations was applied to the test set problem. We believe that while optimizing safety-critical code using LLMs, AEGIS may be the superior approach, as it provides a more comprehensive picture of the optimizations AEGIS was attempting to implement through its retrieved examples.
>
> Additionally, as evidenced by our results on the mean length of AEGIS’s generations, AEGIS’s edits during the initial search steps are much smaller, suggesting that it does not perform extensive edits of the test set program during its initial steps, as compared to RAS, thereby balancing writing performance improving edits while ensuring that edits made in each step are more incremental (and thereby interpretable) in nature.
>
> **References**
>
> 1. Qwen Team. (2025). Qwen3-Coder: Agentic Coding in the World.
> 2. Shypula, A. G., Madaan, A., Zeng, Y., Alon, U., Gardner, J. R., Yang, Y., Hashemi, M., Neubig, G., Ranganathan, P., Bastani, O., & Yazdanbakhsh, A. (2024). Learning Performance-Improving Code Edits. In _The Twelfth International Conference on Learning Representations._
> 3. Huang, D., Qing, Y., Shang, W., Cui, H., & Zhang, J. M. (2024). Effibench: Benchmarking the efficiency of automatically generated code. _Advances in Neural Information Processing Systems, 37,_ 11506-11544.
> 4. Du, M., Luu, A. T., Ji, B., Liu, Q., & Ng, S. K. (2024). Mercury: A code efficiency benchmark for code large language models. _Advances in Neural Information Processing Systems, 37,_ 16601-16622.
> 5. Qiu, R., Zeng, W. W., Ezick, J., Lott, C., & Tong, H. (2025). How efficient is llm-generated code? a rigorous & high-standard benchmark. In _The Thirteenth International Conference on Learning Representations._
> 6. Du, M., Tuan, L. A., Liu, Y., Qing, Y., Huang, D., He, X., Liu, Q., Ma, Z., & Ng, S. K. (2025). Afterburner: Reinforcement learning facilitates self-improving code efficiency optimization. _NeurIPS 2025._

---

### Official Review · Reviewer_Nseg · 2025-10-31

**Soundness:** 2
**Presentation:** 3
**Contribution:** 2
**Rating:** 4
**Confidence:** 3

**Summary:**

This paper addresses the limitation that large language models (LLMs) struggle with "out-of-the-box" program optimization, which proposes two novel retrieval-based adaptation methods: Retrieval Augmented Search (RAS) and Atomic Edit Guided Search (AEGIS).  RAS introduces a beam search mechanism combined with a crucial insight: performing contextual retrieval based on LLM-generated natural language descriptions of programs significantly outperforms code-based retrieval. AEGIS extends this by decomposing training examples into "atomic edits," enabling more incremental and interpretable optimizations. This work makes some contributions by advancing the efficacy and interpretability of LLM-driven program optimization through intelligent retrieval and structured search, offering a compelling framework for future research in code generation tasks.

**Strengths:**

- **Clear Baselines and Ablations**: Comparisons against state-of-the-art dynamic retrieval, "Instruct Only," and "No Contextual" ablations (using source code retrieval instead of contextual retrieval) effectively isolate the impact of core innovations. For example, RAS achieves an 8.61× speedup on PIE—2.04× better than dynamic retrieval—while the "No Contextual" variant's 3.63× speedup confirms that contextual retrieval is essential.
- **Cross-Language Generalizability**: Evaluations on PIE (C++ optimization with deterministic gem5 simulation) and Mercury (Python optimization with runtime percentiles) demonstrate strong cross-language generalizability, addressing a key limitation of prior work—for example, SBLLM only reported C++ results with 1.55× speedup.

**Weaknesses:**

- **Effectiveness of Atomic Operation Decomposition in AEGIS:** The paper proposes AEGIS, which decomposes atomic operations in the training dataset. However, experimental results show that AEGIS underperforms RAS. In the "No Contextual" setting, AEGIS's optimization effect falls below Dynamic Retrieval. This raises questions about whether atomic operation decomposition effectively improves optimization performance. While atomic operations enhance interpretability, the paper lacks statistical data from developers on the practical interpretability gains AEGIS provides. Without user-centered evaluation, it remains unclear whether the increased interpretability truly benefits real-world programming scenarios.
- **Inadequate Exploration of Model Generalizability:** The experiments primarily use GPT-4o and Qwen3-Coder on function-level optimization benchmarks. Given the strong capabilities of models like Claude-4, it might achieve high-level program optimization on simple functions without relying on extensive external knowledge. This absence limits the generalizability analysis of the proposed methods across different LLM architectures.
- **Limited Practical Application Potential:** The benchmarks (PIE and Mercury) focus on competitive programming problems with enumerable optimization patterns (e.g., loop-to-DP conversions), making retrieval-based optimization straightforward. However, repository-level tasks (e.g., swe-pref) require understanding broader code context and architectural constraints—challenges not addressed here. It remains unclear whether RAS and AEGIS generalize to large-scale, real-world software optimization.
- **Lack of Exploration on Long-Thinking Models:** For models with strong reasoning capabilities, such as OpenAI o3 or DeepSeek-R1, which outperform at reasoning tasks. Given that program optimization is a complex reasoning-intensive task, these long-thinking models might offer unique advantages. This gap restricts the comprehensiveness of the study in leveraging the full potential of different LLM types for program optimization.

**Questions:**

1. AEGIS performs worse than RAS, and without contextual retrieval it even falls behind Dynamic Retrieval. Does atomic decomposition itself not improve optimization, or are there other issues (like limited atomic edit diversity)? Also, have you tested with actual developers to see if AEGIS truly improves interpretability, rather than just measuring edit distance?
2. Have the authors tried Claude-4, which is strong at code tasks? If not, why?
3. On advanced reasoning models: Have you tested models with strong reasoning capabilities like OpenAI's o3 or DeepSeek-R1? Do you think their reasoning abilities would further improve performance on Instruct-Only?
4. For repository-level optimization tasks such as SWE-perf, how do you envision obtaining the training corpus (slow-fast pairs) needed for your retrieval-based methods?

---

> ### Author Response · Authors · 2025-12-03
> **Response (1/2)**
>
> Thank you for your feedback and for recognizing that our work demonstrates the generalizability of our approach.
>
> **Practical Interpretability Gains of AEGIS**
>
> A crucial component of the AEGIS dataset that differentiates it from RAS is that each retrieved pair is associated with a generalized edit aimed at improving the performance. As a result, we believe that by identifying the generalized atomic edit associated with each retrieved example, AEGIS provides additional insight into the strategies that led to the performance gain. In contrast, if a user were to try to gain the same visibility into the retrieved examples selected by RAS, they would have to identify (in natural language) the optimizations implemented in the retrieved pair selected at each stage of search, and then examine which of those optimizations was applied to the test set problem. We believe that while optimizing safety-critical code using LLMs, AEGIS may be the superior approach, as it provides a more comprehensive picture of the optimizations AEGIS was attempting to implement through its retrieved examples.
>
> **Model Generalizability and Long-Thinking Models**
>
> The reason behind our choice for Qwen3-Coder was that it is a recently released reasoning model that achieves performance comparable to Claude Sonnet 4 on Terminal Bench, SWE-Bench Verified, and Aider Polyglot, and significantly outperforms Deepseek R1 on the same benchmarks. Given the token expenditure we observed on reasoning models in our problem setting, we feel that evaluating experiments at this scale on Claude would be prohibitively expensive. We are currently planning to evaluate our approach against other recent reasoning models, and while we believe that long-thinking models may perform better than prior models, there is still a significant performance gap between the best retrieval-based systems and existing long-thinking models speedup in the instruct-only setting.
>
> **Limited Practical Applications**
>
> We respectfully disagree that our evaluation has limited practical applications due to our choice of benchmark. Repository-level benchmarks, such as SWE-Perf, require an understanding of the broader code context. However, as a result, improvements in performance can be made in their repository-level “realistic” setting simply by choosing an appropriate agentic framework (for example, we can examine Claude’s performance on SWE-Perf in the oracle setting versus the realistic setting using OpenHands). In SWE-Perf’s oracle setting, where all relevant files are provided to the LLM as context, it is unclear whether LLM failures are due to 1) inability to identify relevant context from the provided files, 2) incorrect understanding of which optimizations are relevant to the problem, 3) incorrect code generation or application of an optimization to the problem. While issues 2 and 3 are often language model-level failures, issue 1 can often be mitigated by designing multi-agent systems equipped with tool use (as can be seen by TRAE’s performance on the SWE-Bench Verified Leaderboard). As a result, we believe it is essential to develop both black-box and post-training methods for mitigating issues 2 and 3, which can later be integrated into agentic systems that address issue 1.
>
> Additionally, we believe that there is widespread recognition that issues 2 and 3 are challenging problems that require dedicated benchmarks within the community. There has been significant interest in recent years in evaluating model performance on issues 2 and 3 independently. Benchmarks such as PIE (ICLR 2024), EffiBench (NeurIPS 2024), Mercury (NeurIPS 2024), ENAMEL (ICLR 2025), and Venus (NeurIPS 2025) all study the optimization of single-file programs and have demonstrated that frontier models still struggle to identify and apply program optimizations that successfully speed up such single-file programs. While several benchmarks have been proposed to study this problem effectively, we have not found a comparable number of works that study approaches to improve model performance on these tasks. Thus, a significant gap exists in the literature regarding methods to improve LLM-guided performance optimization, and our work aims to address this gap.
>
> Finally, we would like to highlight the scope of our work. While SWE-Perf comprises 140 tasks from 9 GitHub repositories, we use benchmarks that both have more instances (973 test-set problems for PIE and 256 for Mercury) and are drawn from a much wider set of sources (CodeNet problems for PIE and LeetCode problems for Mercury). As a result, our methods can evaluate performance for a wide range of algorithmic optimizations and implementations, allowing us to analyze the distribution of failed optimization implementations in Appendix D - tasks that we believe would be challenging in repository-based benchmarks such as SWE-Perf.

---

> ### Author Response · Authors · 2025-12-03
> **Response (2/2)**
>
> **References**
> 1. Qwen Team. (2025). Qwen3-Coder: Agentic Coding in the World.
> 2. Shypula, A. G., Madaan, A., Zeng, Y., Alon, U., Gardner, J. R., Yang, Y., Hashemi, M., Neubig, G., Ranganathan, P., Bastani, O., & Yazdanbakhsh, A. (2024). Learning Performance-Improving Code Edits. In _The Twelfth International Conference on Learning Representations._
> 3. Huang, D., Qing, Y., Shang, W., Cui, H., & Zhang, J. M. (2024). Effibench: Benchmarking the efficiency of automatically generated code. _Advances in Neural Information Processing Systems, 37,_ 11506-11544.
> 4. Du, M., Luu, A. T., Ji, B., Liu, Q., & Ng, S. K. (2024). Mercury: A code efficiency benchmark for code large language models. _Advances in Neural Information Processing Systems, 37,_ 16601-16622.
> 5. Qiu, R., Zeng, W. W., Ezick, J., Lott, C., & Tong, H. (2025). How efficient is llm-generated code? a rigorous & high-standard benchmark. In _The Thirteenth International Conference on Learning Representations._
> 6. Du, M., Tuan, L. A., Liu, Y., Qing, Y., Huang, D., He, X., Liu, Q., Ma, Z., & Ng, S. K. (2025). Afterburner: Reinforcement learning facilitates self-improving code efficiency optimization. _NeurIPS 2025._

---

### Note · Authors · 2025-12-03

I have read and agree with the venue's withdrawal policy on behalf of myself and my co-authors.